# Dietary Arachidonic Acid (20:4n-6) Levels and Its Effect on Growth Performance, Fatty Acid Profile, Gene Expression for Lipid Metabolism, and Health Status of Juvenile California Yellowtail (*Seriola dorsalis*)

Bruno Cavalheiro Araújo [1,2], Arleta Krystyna Skrzynska [3,4], Victor Hugo Marques [5], Aurora Tinajero [3], Oscar Basílio Del Rio-Zaragoza [3], Maria Teresa Viana [3] and José Antonio Mata-Sotres [6,7,*]

1 Cawthron Institute, Nelson 7010, New Zealand; bruno.araujo@cawthron.org.nz
2 Núcleo Integrado de Biotecnologia, Universidade de Mogi das Cruzes, Mogi das Cruzes 08780-911, Brazil
3 Instituto de Investigaciones Oceanológicas, Universidad Autónoma de Baja California (UABC), Ensenada 21100, Mexico; arleta.skrzynska@uca.es (A.K.S.); tinajero.aurora@uabc.edu.mx (A.T.); oscar.delrio@uabc.edu.mx (O.B.D.R.-Z.); mtviana@hotmail.com (M.T.V.)
4 Department of Biology, Faculty of Marine and Environmental Sciences, Campus de Excelencia Internacional del Mar (CEI-MAR), University of Cádiz, 11510 Puerto Real, Spain
5 Departamento de Fisiologia, Instituto de Biociências, Universidade de São Paulo, São Paulo 05508-070, Brazil; victor.marqueslorenti@gmail.com
6 CONACYT, Instituto de Investigaciones Oceanológicas, UABC, Ensenada 21100, Mexico
7 Departamento el Hombre y su Ambiente, Universidad Autonoma Metropolitana, Unidad Xochimilco, Ciudad de México 52919, Mexico
* Correspondence: jmata@correo.xoc.uam.mx

**Abstract:** Arachidonic acid (ARA, 20:4n-6) fed to *Seriola dorsalis* juveniles at different levels was evaluated. After a seven-week feeding trial, growth performance, hepatopancreas and muscle fatty acid (FA) composition, expression of lipid-relevant genes, and blood parameters were evaluated. Four isoproteic and isolipidic experimental diets were formulated to contain 44% crude protein and 11% lipids with graded inclusion levels of ARA, 0% (Control), 0.4, 0.9, and 1.4% of the total diet. *S. dorsalis* juveniles (14.54 ± 0.18 g) were randomly divided into twelve tanks with fifteen animals each. The animals were hand fed three times per day to apparent satiation. Dietary treatments did not significantly affect the growth performance, SGR, FCR, and feed intake of fish. Different levels of ARA in the experimental diets directly influenced liver and muscle FA profiles, with significant changes in ARA and EPA deposition between Control treatment and 1.4%, in both tissues. The expression of arachidonate 5-lipoxygenase (*alox5*), acyl-CoA dehydrogenase very long chain (*acadvl*), carnitine O-palmitoyltransferase 1(*cpt1a*) was significantly affected by dietary treatments, with an expression increasing accordingly to the increasing ARA levels. In contrast, a reduction of fatty acid synthase (*fas*) and proliferator-activated receptor alpha (*ppara*) expression was significantly reduced as ARA increased in the diet. In addition, a significant reduction in blood cortisol and glucose was found at a 0.9% ARA level compared to the other treatments. Based on the performance, cortisol levels, the gene expression for eicosanoids synthesis, and lipid metabolic pathways, the present study suggests a maximum ARA inclusion of 0.9% in diets for California yellowtail juveniles, *S. dorsalis*.

**Keywords:** stress response; gene expression; lipid replacement; aquafeed; marine fish

## 1. Introduction

Carnivorous marine fish are highly fish oil (FO) dependent, mainly due to the nutritional requirement for long-chain polyunsaturated fatty acids (LC-PUFA), especially those from the n-3 series. Docosahexaenoic acid (DHA, 22:6n-3) and eicosapentaenoic acid (EPA, 20:5n-3) are the most important LC-PUFA for marine species [1–4]. Unlike freshwater and euryhaline species, most marine fish are unable to bioconvert or synthesize



biologically meaningful quantities of EPA and DHA, such as other physiologically active LC-PUFA from the n-6 series as arachidonic acid (ARA, 20:4n-6) [5–8]. This metabolic insufficiency has been attributed to the limited activity of the fatty acid elongase and desaturase enzymes [6,9,10]. Therefore, marine fish rely on dietary sources to meet physiological demand for these bioactive fatty acids (FA) [4,7]. Previous studies showed that entirely FO replacement by alternative lipid sources, mainly from terrestrial vegetables, negatively impacted the performance and muscle FA composition in marine fish and shrimp, mainly due to a reduction of LC-PUFA and inversely increased of monounsaturated (MUFA) and C18 polyunsaturated (C18 PUFA) fatty acids levels [11–14].

Compared to the n-3 LC-PUFA, ARA nutrition studies in marine organisms are scarce, probably due to the low level present in most FO [5]. However, few studies carried out with fish and shrimp species presented promisors results in growth performance [15–18], survival, and stress resistance [19–21], disease resistance [22,23] and reproduction [22,24]. It has been suggested that increased dietary ARA positively affects cortisol levels and stress response in marine fishes [10,23–27]. However, an excess in ARA levels can induce adverse effects in development [28–30]. Another important functional role of the LC-PUFA, especially EPA and ARA, is the substrate to the eicosanoid synthesis by the action of some specific enzymes [5,21,31]. For example, the arachidonate 5-lipoxygenase (*alox5*) enzyme plays a vital role in leukotriene biosynthesis from ARA. Leukotrienes are a family of important mediators of inflammatory processes, mucus secretion, leukocyte aggregation, and superoxide generation in neutrophils [32,33]. Thus, an adequate ARA/EPA ratio in diets to marine fish is essential for the proper functioning of the immune system [5].

California Yellowtail is considered one of the best marine fish species for aquaculture mainly due to its excellent production performance, resistance to different culture conditions, and excellent meat quality (primarily related to the fillet color, texture, and flavor), constantly reaching good market value [34]. However, such as for several other commercial fish species, nutrition can be considered the limiting point to improving profitability. Due to the relevance of ARA to the physiology and, consequently, the performance of the marine fish species, this study aimed to investigate the effect of different inclusion levels. Moreover, the benefits on growth performance, muscle and liver FA profile, expression of lipid metabolism, immune genes, and stress parameters in the blood of *S. dorsalis* were evaluated. These are relevant information to identify optimal levels of inclusion of ARA aiming to achieve efficient FO-free diets for California Yellowtail.

## 2. Materials and Methods

### 2.1. Experimental Design and Fish Handling

One hundred and eighty juveniles of *S. dorsalis* (initial weight of $14.54 \pm 0.18$ g) were kindly donated from a commercial marine finfish hatchery (Maricultivos Baja Sel, S.A. de C.V., Ensenada, B.C., Mexico), and were acclimated for two weeks and randomly stocked (6 fish per tank) into a recirculating aquaculture system (RAS). The experimental system was equipped with 12 tanks of 500 L each (15 fish per tank), supplemental aeration, and a biofilter. Water temperature was controlled and maintained steady with a heater, whereas natural conditions determined the photoperiod. Temperature, dissolved oxygen, and salinity were measured daily using a YSI-55 optical multi-probe meter (YSI Inc., Yellow Springs, OH, USA). In addition, total ammonia, nitrite, and nitrate-nitrogen were quantified three times per week using API test kits (Mars Fishcare Inc, Chalfont, PA, USA). While pH was monitored weekly using a Thermo Scientific Orion 4-Star pH meter (Thermo Scientific, Waltham, MA, USA).

Water quality parameters were maintained within suitable ranges for this species: temperature: $21.5 \pm 0.5$ °C, salinity: $34.1 \pm 0.2$ PSU, dissolved oxygen: $7.15 \pm 0.14$ mg L$^{-1}$, total ammonia nitrogen: $0.28 \pm 0.08$ mg L$^{-1}$, nitrite-nitrogen: $0.97 \pm 0.97$ mg L$^{-1}$, nitrate-nitrogen: $46.7 \pm 30.5$ mg L$^{-1}$, and pH = $7.50 \pm 0.08$ (mean $\pm$ SE). Dietary treatments were randomly assigned to three replicate tanks. Fish were hand-fed three times daily to apparent satiation. All tanks were siphoned daily for the feed intake calculation.

## 2.2. Experimental Diets

The diet formulation, proximal composition, and FA profile (from the LC-PUFA rich oils and experimental diets) are presented in Tables 1 and 2. Four isoproteic and isolipidic experimental diets were formulated to contain 44% crude protein and 11% lipids. The fishmeal (FM) and poultry by-product meal (PBM) were previously defatted using hexane (3:1, hexane:meal). Therefore, FO was used as the primary lipid source in the Control diet, with the inclusion of DHA-rich-oil in the other experimental treatments to reach the requirements of the species. The experimental diets were formulated using a lipid blend containing beef tallow and different ARA-concentrated oils inclusion levels (0, 0.4, 0.9, and 1.4%).

**Table 1.** Ingredient formulation (g kg$^{-1}$) and proximate composition of diets used to fed *S. dorsalis* containing increasing levels of ARA. The dietary formulations are presented as fed-basis and proximate composition in g kg$^{-1}$ on a dry matter basis.

| | Experimental Diets | | | |
|---|---|---|---|---|
| **Ingredients (g kg$^{-1}$ DM)** | **Control** | **0.4%** | **0.9%** | **1.4%** |
| Fish meal [a] | 140 | 140 | 140 | 140 |
| Poultry meal [b] | 360 | 360 | 360 | 360 |
| Wheat meal [c] | 8 | 0 | 0 | 0 |
| Gelatin [d] | 60 | 60 | 60 | 60 |
| Beef tallow [e] | 25 | 94 | 94 | 89 |
| Cholesterol [f] | 0 | 3 | 3 | 3 |
| Starch (Maizena$^{TM}$) | 273 | 265 | 260 | 260 |
| Taurine [g] | 10 | 10 | 10 | 10 |
| Rovimix [h] | 30 | 30 | 30 | 30 |
| Stay C | 1 | 1 | 1 | 1 |
| DHANatur [i] | 0 | 30 | 30 | 30 |
| Fish oil [j] | 90 | 0 | 0 | 0 |
| ARA-enriched oil [k] | 0 | 4 | 9 | 14 |
| Sodium Benzonate | 2 | 2 | 2 | 2 |
| Choline chloride | 0.9 | 0.9 | 0.9 | 0.9 |
| Tocopherol | 0.1 | 0.1 | 0.1 | 0.1 |
| **Proximate Composition (g kg$^{-1}$)** | | | | |
| Protein | 449.3 | 449.4 | 468.7 | 461.9 |
| Lipid | 109.2 | 122.5 | 113.5 | 112.5 |
| Dry Matter | 95.07 | 91.79 | 92.93 | 97.10 |
| Ash | 80.7 | 81.8 | 85.0 | 83.1 |
| NFE | 360.8 | 346.3 | 332.8 | 342.5 |

Nitrogen-free extract; NFE (%) = 100 − (% crude protein + % total lipid + % ash). NFE includes fibers. [a] Sardine fishmeal from Californias Bahía SA de CV, Ensenada BC, Mexico. [b] Pet food grade from NARA, USA. [c] Molinera del Valle S.A. de C.V. Mexicali, Baja California, México. [d] Progel Mexicana SA de CV, Leon, Guanajuato, México. [e] Industrial de Grasas y Derivados, Tijuana BC, México. [f] Cholesterol, MITSUI, México. [g] Taurine Future foods, Tlanepanla, Estado de México, México [h] DSM Nutritional Products Mexico SA de CV, Guadalajara, Jalisco, Mexico, contains in g kg ρ-aminobenzoic acid 1.45; biotin 0.02; myo-inositol 14.5; nicotinic acid 2.9; Capantothenate 1.0; pyridoxine-HCl 0.17; riboflavin 0.73; thiamine-HCl 0.22; menadione 0.17; α-tocopherol 1.45; cyanocobalamine 0.0003; calciferol 0.03; L-ascorbyl-2- phosphate-Mg 0.25; folic acid 0.05; choline chloride 29.65; retinol 0.015; NaCl 1.838; MgSO$_4$ 7H$_2$O 6.85; NaH$_2$PO$_4$ 2H$_2$O 4.36; KH$_2$PO$_4$ 11.99; Ca (H$_2$PO$_4$)$_2$ 2H$_2$O 6.79; Fe-citrate 1.48; Ca-lactate 16.35; AlCl$_3$ 6H$_2$O 0.009; ZnSO$_4$ 7H$_2$O 0.17; CuCl$_2$ 0.0005;.MnSO$_4$ 4H$_2$O 0.04; KI 0.008; CoCl$_2$ 0.05 and Stay–C (Vitamin C). [i] ADM, Archer Daniels Midland Company, Mexico. [j] Scoular de México, S. de R.L. de C.V. [k] Fungal ARA concentrated oil (>40% ARA). Jiangsu Tiankai Biotechnology Co., Ltd. Nanjing. China.

ΣSFA, ΣMUFA, ΣPUFA, Σn-3 PUFA, and Σn-6 PUFA are the sum of saturated, mononsaturated, polyunsaturated, n-3 polyunsaturated, and n-6 polyunsaturated, respectively. EPA/ARA is the ratio between EPA and ARA; DHA/ARA is the ratio between DHA and ARA.

All experimental diets were manufactured at the LINDEAACUA plant at the II0-UABC facilities (Ensenada, Mexico). Briefly, the macronutrients were pulverized to 0.5 mm (Inmimex M-300, Santa Justina Ecatepec, CDMX, Mexico) and sifted (Kemutek-Gardner K300, Bristol, TN, USA). After that, the dough was mixed using a vertical cutter-mixer

(Robot Coupe R-60, Ridgeland, AL, USA) to obtain a homogeneous mass. Then, the micronutrients were incorporated into the bulk meal. At the same time, the oil blend of each experimental diet was added and mixed thoroughly. Finally, water was added until achieved the desired texture. The mixed diets (Robot-Coupe, modelR10, Ridgeland, AL, USA) were pelleted at 5 mm in a meat grinder (M32–5, Tor-Rey, Ridgeland, AL, Mexico) and dried at 60 °C in a forced air oven for 24 h. Once dried, diets were kept cooled (4 °C) throughout the feeding trial.

**Table 2.** Fatty acid composition of the experimental diets and oils (% total fatty acids).

| | | | Experimental Diets | | |
|---|---|---|---|---|---|
| **Fatty Acid** | **ARA Oil** | **Control** | **0.4%** | **0.9%** | **1.4%** |
| 14:0 | 2.29 | 6.87 | 6.83 | 5.73 | 6.15 |
| 16:0 | 8.96 | 23.51 | 19.58 | 20.10 | 19.75 |
| 18:0 | 9.70 | 9.50 | 12.71 | 13.00 | 12.63 |
| ∑SFA | 20.95 | 39.88 | 39.13 | 38.84 | 38.53 |
| 16:1n-7 | 0.11 | 7.35 | 3.05 | 3.03 | 2.83 |
| 18:1n-7 | 1.46 | 3.25 | 1.87 | 1.81 | 1.71 |
| 18:1n-9 | 28.02 | 22.70 | 28.17 | 29.18 | 28.20 |
| ∑MUFA | 29.59 | 33.30 | 33.09 | 34.01 | 32.74 |
| 18:2n-6 | n.d. | 4.74 | 6.28 | 6.96 | 7.38 |
| **20:4n-6 (ARA)** | **44.95** | **2.39** | **2.21** | **3.49** | **4.55** |
| ∑n-6 PUFA | 44.95 | 7.13 | 8.49 | 10.45 | 11.93 |
| 18:3n-3 | n.d. | 0.80 | 1.72 | 1.14 | 0.84 |
| 18:4n-3 | n.d. | 1.07 | 1.42 | 1.00 | 1.18 |
| 20:5n-3 | n.d. | 3.35 | 1.64 | 1.84 | 1.76 |
| 22:5n-3 | n.d. | 4.25 | 3.19 | 2.58 | 3.96 |
| 22:6n-3 | n.d. | 4.75 | 5.64 | 5.26 | 4.56 |
| ∑n-3 PUFA | n.d. | 14.21 | 13.61 | 11.80 | 12.29 |
| ∑PUFA | 44.95 | 21.35 | 22.10 | 22.25 | 24.22 |
| EPA/ARA | - | 1.40 | 0.74 | 0.74 | 0.39 |
| DHA/ARA | - | 1.98 | 2.55 | 1.55 | 1.0 |
| Others | 4.51 | 5.48 | 5.68 | 4.90 | 4.50 |

*2.3. Sampling*

After a 50-day feeding trial, fish were counted and group-weighed by the tank to assess performance in terms of the following metrics:

Specific growth rate (SGR, %d) = 100 × (ln final weight–ln initial weight) × number of days).

Feed Intake = (FI, % $day^{-1}$) = 100 × (total amount of the feed consumed/(initial body weight + final body weight)/2)/days).

FCR (Feed Conversion Ratio) = total feed consumed/wet weight gained.

Condition Factor (CF) = (final body weight/body $length^3$) × 100.

Hepatosomatic index (HSI, %) = (hepatopancreas weight/body weight) × 100.

Viscerosomatic index (VSI, %) = (viscera weight/body weight) × 100.

Three fish per tank were euthanized according to the UABC (Universidad Autónoma de Baja California) protocols on using animals for experimental purposes with 2-phenoxyl-ethanol solution (100 ppm); immediately, blood samples were collected from caudal vasculature using regular syringes. Blood aliquots were placed into tubes without anticoagulants and then centrifuged for 10 min and used for blood chemistry. All serum samples were stored at –20 °C for further analyses. Then the same fish were weighed and dissected to collect the morphometric parameters.

*2.4. Proximate Composition and Fatty Acid Profile*

All experimental diets were analyzed in triplicate, according to AOAC [35]. Dry weight and the ash content of the diets were determined by drying ground samples at 60 °C for 24 h, followed by carbonization in a muffle furnace at 550 °C for six hours. Crude

protein was analyzed by the micro-Kjeldahl method (UDK 129, Velp, Italy), and the content was calculated by nitrogen conversion (%N × 6.25). First, lipid analysis was performed by the Soxhlet method according to the AOAC [35], using petroleum ether as a carrier. Next, the lipid profile was analyzed using methanol/dichloromethane, initiating a lipid extraction from the fresh sample according to Folch et al. [36]. After that, the lipid extract was methylated following the transmethylation method described by Parrish et al. [37]. Next, fatty acids methyl esters (FAMEs) were analyzed using gas chromatography equipped with a flame ionization detector (Agilent GC 6880, Agilent Technologies, Santa Clara, CA, USA) using hydrogen as the carrier gas. The GC column (60 m × 0.25 mm with 0.25 μm film thickness; Agilent 122-2362 dB-23) conditions were: oven temperature initial of 50 °C for 1 min, 50 to 140 °C at 30 °C min$^{-1}$, held at 140 °C for 5 min, from 140 to 240 °C at 4 °C min$^{-1}$, and finally 240 °C for 20 min. The injector and detector temperatures were kept at 230 and 260 °C, respectively. Finally, FAMEs were identified and quantified, comparing retention times against an internal standard (37 Component FAME mix, PUFA 1 and PUFA 3, Supelco/Sigma-Aldrich, St. Louis, MO, USA).

*2.5. RNA Extraction and Quantitative PCR*

Liver tissue samples preserved in RNAlater (Ambion) were individually processed for total RNA extraction using the PureLink® RNAlink Minikit (Ambion). Genomic DNA (gDNA) was removed via on-column using PureLink® DNase (Invitrogen) following the manufacturer's instruction. A micropistill was used to homogenize the tissue before the extraction. The quantity and quality of RNA were measured using gel electrophoresis and a spectrophotometer (Nanodrop® LITE, Thermo Fisher Scientific INC., Wilmington, USA). Only RNA samples with OD260nm-OD280nm ratios between 1.90 and 2.10 were used for expression quantification.

Total RNA (500 ng) was reverse-transcribed in a 20 μL reaction using the High-Capacity cDNA Reverse Transcription kit (Applied Biosystems; Carlsbad, CA, USA) in a Verity 96 well thermal cycler (Applied Biosystems). The reverse transcription program consisted of 10 min at 25 °C, 120 min at 37 °C, 5 min at 85 °C, and finally kept at 4 °C. qPCR reactions were performed with one ng of cDNA, sense, and antisense primers (200 nM each, indicated in Table 3) and SYBR® Select Master Mix (Applied Biosystems). In addition, reactions were conducted in 10 μL in MicroAmp® Fast Optical 96-well reaction plates (Applied Biosystems) covered with MicroAmp® Optical Adhesive Film (Applied Biosystems).

**Table 3.** Primers pairs used for q-PCR Primer sequences, amplicon sizes in base pairs (bp), reaction efficiencies (E), and Pearson's coefficients of determination ($R^2$) are indicated.

| Gene (Symbol) | Fwd Sequence (5′-3′) | Rev Sequence (5′-3′) | Size (bp) | E | $R^2$ |
|---|---|---|---|---|---|
| *actb* | TGCGTGACATCAAGGAGAAG | AGGAAGGAAGGCTGGAAGAG | 175 | 1.00 | 0.99 |
| *acadvl* | ATTTGGGGTTCAGTGTCTCG | CTGTGACGACAAAAGCCAGA | 153 | 1.13 | 0.99 |
| *alox5* | ACAAAACCTCGCTGCAGACT | CTGTGCCCACCAGTGTAATG | 187 | 1.02 | 0.96 |
| *cpt1a* | CCATCATGGTCAACAGCAAC | ACGTTCGTATTGGGATGAGC | 188 | 1.11 | 0.99 |
| *elovl* | TTACTGCTGTGTGGCATGGT | CTGGCATGGTGGTAGATGTG | 196 | 1.02 | 0.92 |
| *fas* | CCTGCTGGCTTTAGAAAACG | ACGGCAGTATCCATTTCCTG | 181 | 1.02 | 0.92 |
| *ppara* | CAGCCACAAGACTCTGGTCA | TCTCGTGCTCCAGAGAGTCA | 200 | 0.96 | 0.99 |
| *igf1* | TCTTCAAGAGTGCGATGTGC | GGCCATAGCCTGTTGGTTTA | 189 | 0.99 | 0.99 |

Relative gene quantification was calculated by the $\Delta\Delta C_T$ method [38], and relative gene quantification was calculated using an automated threshold and walking baseline to determine the $C_T$ values. PCR conditions were: an initial denaturation and polymerase activation step for 10 min at 95 °C; 40 cycles of denaturing for 15 s at 95 °C, annealing and extension for 45 s at 60 °C; and a final melting curve from 60 °C to 95 °C for 20 min to check for primer-dimer artifacts. The qPCR optimization conditions were made on primer annealing temperature (60 °C), primer concentration (200 nM), and template concentration (five 1:10 dilution series in triplicate from 10 ng to 1 pg of input RNA). *β-actin* was used as

the internal reference gene (GenBank acc. no KT229636.1). GenBank accession numbers for the studied genes are: XM_023420820.1 for acyl-CoA dehydrogenase very long chain (*acadvl*), XM_023405784.1 arachidonate 5-lipoxygenase (*alox5*), XM_023415436.1 for carnitine O-palmitoyltransferase 1 (*cptla*), GU047382.1 for fatty acid elongase (*elovl*), KT895233.1 for fatty acid synthase (*fas*), XM_023402033.1 for peroxisome proliferator-activated receptor (*ppara*) and AB439208.1 for insulin-like growth factor 1 (*igf1*).

### 2.6. Cortisol, Glucose, and Total Protein Serum Levels Quantification

Cortisol was determined using enzyme-linked immunosorbent assay kits, reading at 450 nm following the manufacturer's instructions (MexLab Group, Jalisco, Mexico). Glucose and total protein were determined using a colorimetric kits assay (MexLab Group, Jalisco, Mexico), following the manufacturer's instructions. Glucose was analyzed with the glucose oxidase method (God-Pap) at 505 nm. While the total soluble protein was analyzed using the Biuret method at 540 nm and reported as bovine albumin serum (BSA) equivalent using a microplate reader (Multiskan GO, Thermo Scientific).

### 2.7. Statistical Analysis

Normality and homogeneity of variance were tested for all analyzed parameters. When the data allowed a parametric analysis, the comparisons between different treatments were performed by one-way analysis of variance (ANOVA) followed by Tukey's HSD test. For all cases, statistical significance was set at $p < 0.05$. Statistical analysis was performed using the software STATISTICA 8.0™ (StatSoft, Inc., Tulsa, OK, USA).

## 3. Results

### 3.1. Performance and Biological Index

The performance and biological indexes of fish according to different ARA treatments are presented in Table 4. Even if no significant differences were revealed, low final weight was observed when ARA was included in the diet at the lowest level (0.4%), with a tendency to increase with higher inclusion levels of ARA (0.9 and 1.4%). Additionally, there was no effect on the SGR and FCR among the ARA levels (Table 4). On the other hand, the CF was directly affected by dietary ARA inclusion. The fish of Control treatment revealed a significant ($p < 0.05$) increase, with a trend to decrease with a reduction in ARA level. There were no significant differences in HSI and VSI among the different ARA treatments. However, the lowest values were observed in the fish from the treatments corresponding to the lower inclusion levels (0.4 and 0.9%).

**Table 4.** Growth performance of *S. dorsalis* fed diets containing different levels of ARA.

| | Control | 0.4% | 0.9% | 1.4% | PSE | ANOVA *p*-Value |
|---|---|---|---|---|---|---|
| Initial weight (g) | 14.71 ± 0.13 | 14.54 ± 0.12 | 14.29 ± 0.07 | 14.62 ± 0.17 | 0.07 | 0.18 |
| Final weight (g) | 41.54 ± 1.34 | 37.79 ± 0.48 | 41.07 ± 0.48 | 41.28 ± 2.68 | 0.79 | 0.32 |
| [a] FI (% day$^{-1}$) | 2.33 ± 0.16 | 2.47 ± 0.07 | 2.25 ± 0.15 | 2.38 ± 0.09 | 0.06 | 0.51 |
| [b] SGR (% day$^{-1}$) | 2.15 ± 0.12 | 1.91 ± 0.02 | 2.11 ± 0.02 | 2.07 ± 0.13 | 0.05 | 0.32 |
| [c] FCR | 1.23 ± 0.11 | 1.39 ± 0.05 | 1.16 ± 0.07 | 1.26 ± 0.10 | 0.05 | 0.35 |
| [d] CF | 3.10 ± 0.16 [a] | 1.43 ± 0.06 [b] | 1.63 ± 0.02 [b] | 1.76 ± 0.14 [b] | 0.20 | 0.01 |
| [e] HSI% | 1.20 ± 0.23 | 1.05 ± 0.06 | 1.04 ± 0.04 | 1.21 ± 0.09 | 0.06 | 0.70 |
| [f] VSI% | 10.90 ± 2.01 | 8.23 ± 0.47 | 8.78 ± 0.13 | 9.23 ± 0.18 | 0.54 | 0.35 |
| Survival (%) | 95.5 ± 0.36 | 95.5 ± 1.25 | 97.7 ± 1.32 | 95.5 ± 0.89 | 1.29 | 0.93 |

Values are presented as means ± SE. Moreover, pooled standard error (PSE) is given from three replicates per treatment and *p* values resulting from a one-way ANOVA test are also provided. Different letters indices represent significantly different values ($p < 0.05$) within the same row. [a] Feed intake, [b] specific growth rate, [c] feed conversion rate, [d] condition factor, [e] hepatosomatic index, [f] viscerasomatic index.

### 3.2. Fatty Acid Composition of Tissues

The total FA compositions of liver and muscle are presented in Tables 5 and 6. In general, different ARA levels included in the experimental diets directly influence the FA profile of both tissues.

**Table 5.** Liver fatty acid composition (% of total FA) of *S. dorsalis* juvenile fed with different ARA concentrations.

| | Treatments | | | | |
|---|---|---|---|---|---|
| **Fatty Acids** | **Control** | **0.4%** | **0.9%** | **1.4%** | ***p* Value** |
| 14:0 | 2.81 ± 0.18 | 3.43 ± 0.04 | 3.08 ± 0.40 | 2.94 ± 0.20 | 0.37 |
| 16:0 | 19.72 ± 0.97 [a] | 17.21 ± 0.18 [ab] | 17.06 ± 0.66 [ab] | 16.13 ± 0.23 [b] | 0.01 |
| 18:0 | 7.71 ± 0.39 | 7.39 ± 0.11 | 7.56 ± 0.21 | 7.39 ± 0.15 | 0.75 |
| 20:0 | 0.45 ± 0.03 | 0.45 ± 0.03 | 0.68 ± 0.17 | 0.41 ± 0.01 | 0.20 |
| ∑SFA | 30.69 ± 1.57 [a] | 28.47 ± 0.37 [ab] | 28.38 ± 1.45 [ab] | 26.88 ± 0.58 [b] | 0.01 |
| 16:1n-7 | 3.47 ± 0.69 | 2.75 ± 0.15 | 2.90 ± 0.74 | 2.36 ± 0.21 | 0.54 |
| 18:1n-7 | 2.36 ± 0.34 | 1.88 ± 0.09 | 2.28 ± 0.48 | 2.22 ± 0.50 | 0.82 |
| 18:1n-9 | 17.94 ± 3.65 | 29.02 ± 1.52 | 22.63 ± 2.18 | 25.12 ± 1.86 | 0.06 |
| 24:1n-9 | 1.18 ± 0.31 | 0.58 ± 0.09 | 0.71 ± 0.12 | 0.62 ± 0.11 | 0.15 |
| ∑MUFA | 24.95 ± 5.00 | 34.22 ± 1.89 | 28.52 ± 3.53 | 30.32 ± 2.67 | 0.06 |
| 18:2n-6 | 7.70 ± 0.83 | 9.68 ± 0.22 | 8.80 ± 0.60 | 9.80 ± 0.54 | 0.11 |
| 20:2n-6 | 0.43 ± 0.02 | 0.52 ± 0.04 | 0.94 ± 0.30 | 0.57 ± 0.02 | 0.16 |
| 20:4n-6 | 7.07 ± 1.70 [b] | 6.56 ± 0.51 [b] | 9.39 ± 1.18 [ab] | 12.37 ± 0.59 [a] | 0.02 |
| ∑n-6 PUFA | 15.19 ± 2.53 [b] | 16.76 ± 0.47 [ab] | 19.12 ± 1.35 [ab] | 22.74 ± 0.36 [b] | 0.03 |
| 18:3n-3 | 0.57 ± 0.04 | 0.51 ± 0.01 | 0.73 ± 0.23 | 0.51 ± 0.02 | 0.56 |
| 20:5n-3 | 8.85 ± 0.10 [a] | 1.09 ± 0.32 [b] | 0.52 ± 0.02 [b] | 0.46 ± 0.04 [b] | 0.01 |
| 22:5n-3 | 1.71 ± 0.04 | 2.77 ± 0.18 | 2.47 ± 0.42 | 2.73 ± 0.27 | 0.21 |
| 22:6n-3 | 15.61 ± 1.33 | 12.37 ± 0.87 | 14.00 ± 1.19 | 12.32 ± 1.42 | 0.25 |
| ∑n-3 PUFA | 26.75 ± 0.90 [a] | 16.74 ± 1.03 [b] | 17.72 ± 1.36 [b] | 16.03 ± 1.71 [b] | 0.01 |
| ∑PUFA | 39.16 ± 1.28 | 33.50 ± 1.41 | 38.51 ± 1.36 | 38.77 ± 1.87 | 0.84 |
| Others | 5.20 ± 0.51 | 3.81 ± 0.12 | 4.59 ± 0.35 | 4.03 ± 0.45 | 0.12 |

Values represent means ± standard deviation (*n* = 3). ∑SFA, ∑MUFA, ∑PUFA, ∑n-3 PUFA, and ∑n-6 PUFA are the sum of saturated, monounsaturated, polyunsaturated, n-3 polyunsaturated, and n-6 polyunsaturated, respectively. [ab] Different letters indicate statistical differences between experimental diets, by Tukey's test (*p* < 0.05).

**Table 6.** Muscle fatty acid composition (% of total FA) of *S. dorsalis* juvenile fed with different ARA concentrations.

| | Treatments | | | | |
|---|---|---|---|---|---|
| **Fatty Acid** | **Control** | **0.4%** | **0.9%** | **1.4%** | ***p* Value** |
| 14:0 | 1.26 ± 0.03 | 1.48 ± 0.10 | 2.02 ± 0.05 | 2.29 ±0.73 | 0.25 |
| 16:0 | 15.34 ± 0.46 | 14.07 ± 0.64 | 14.94 ± 0.60 | 14.05 ±0.13 | 0.19 |
| 18:0 | 8.07 ± 0.39 | 9.08 ± 0.50 | 9.11 ± 0.04 | 8.42 ±0.44 | 0.23 |
| 20:0 | 0.80 ± 0.27 | 0.54 ± 0.14 | 0.35 ± 0.13 | 0.42 ± 0.09 | 0.31 |
| ∑SFA | 16.59 ± 0.49 | 15.56 ± 0.58 | 16.96 ± 0.54 | 16.34 ±0.79 | 0.42 |
| 16:1 | 2.50 ± 0.04 | 1.65 ± 0.02 | 2.11 ± 0.21 | 2.37 ±0.35 | 0.06 |
| 18:1n-7 | 3.24 ± 0.21 [a] | 2.24 ± 0.06 [b] | 2.20 ± 0.05 [b] | 2.09 ± 0.04 [b] | 0.01 |
| 18:1n-9 | 11.79 ± 1.05 [b] | 17.99 ± 0.97 [ab] | 22.35 ± 2.96 [a] | 22.77 ±2.81 [a] | 0.01 |
| ∑MUFA | 25.61 ± 3.45 | 30.97 ± 1.41 | 35.78 ± 2.84 | 35.66 ±2.75 | 0.35 |
| 18:2n-6 | 5.76 ± 0.01 [b] | 8.42 ± 0.10 [a] | 8.87 ± 0.10 [a] | 8.74 ±0.61 [a] | 0.01 |
| 18:3n-3 | 0.78 ± 0.18 | 0.74 ± 0.20 | 0.48 ± 0.07 | 0.58 ±0.08 | 0.49 |
| 20:2n-6 | 1.35 ± 0.25 | 0.99 ± 0.33 | 0.56 ± 0.09 | 0.55 ±0.03 | 0.07 |
| 20:4n-6 | 4.12 ± 0.21 [a] | 2.46 ± 0.49 [b] | 1.23 ± 0.24 [bc] | 0.98 ±0.02 [c] | 0.01 |
| 20:5n-3 | 8.78 ± 0.72 [ab] | 6.12 ± 0.13 [a] | 8.78 ± 0.46 [ab] | 9.72 ±1.05 [b] | 0.02 |
| 22:5n-3 | 0.78 ± 0.17 | 0.56 ± 0.13 | 0.28 ± 0.06 | 0.41 ±0.03 | 0.06 |
| 22:6n-3 | 23.25 ± 0.25 [a] | 20.22 ± 0.93 [b] | 17.92 ± 0.95 [bc] | 14.55 ±1.58 [c] | 0.01 |
| 24:1n-9 | 2.97 ± 0.82 [a] | 1.42 ± 0.06 [ab] | 0.87 ± 0.16 [b] | 1.19 ±0.18 [ab] | 0.03 |
| ∑n-6 PUFA | 11.24 ± 0.45 | 11.88 ± 0.77 | 10.66 ± 0.23 | 10.28 ±0.65 | 0.27 |
| ∑n-3 PUFA | 33.59 ± 1.32 [a] | 27.65 ± 0.51 [ab] | 27.48 ± 1.57 [ab] | 25.27 ±2.56 [b] | 0.02 |
| ∑PUFA | 44.83± 1.77 [a] | 39.53 ± 0.29 [ab] | 38.15 ± 1.80 [ab] | 35.55 ±2.14 [b] | 0.01 |
| Others | 9.20 ±1.06 | 11.95 ± 1.58 | 7.85 ± 1.28 | 10.81 ±3.10 | 0.52 |

Values represent means ± standard deviation (*n* = 3). ∑SFA, ∑MUFA, ∑PUFA, ∑n-3 PUFA, and ∑n-6 PUFA are the sum of saturated, monounsaturated, polyunsaturated, n-3 polyunsaturated, and n-6 polyunsaturated, respectively. [abc] Different letters indicate statistical differences between experimental diets, by Tukey's test (*p* < 0.05).

Concerning the FA deposition in the liver (Table 5), the total saturated fatty acids (SFA) were significantly lower as the inclusion of ARA increased in the experimental diets. Only the palmitic acid (16:0) resulted in significant differences, following a reduction trend toward ARA was enriched. Fish fed with 1.4% dietary treatment resulted in significantly higher ARA deposition in the liver than the other experimental treatments, with lower ARA levels (Control and 0.4%). The opposite was revealed with EPA, where a significant increase was found in Control fish than in other treatments enriched with ARA. Further, no significant differences were observed in DHA in liver tissues. Meanwhile, total n-3 PUFA was significantly reduced at any ARA inclusion level (0.4, 0.9, and 1.4%) compared to the Control treatment. On the other hand, a gradual increase was observed in the n-6 PUFA, influenced mainly by the ARA content in the tissues.

In the muscle (Table 6), no significant differences were observed in the percentage of total monounsaturated fatty acid (MUFA), 18:1n-7 decreased along with ARA increase in the diets, and a substantial accumulation of 18:1n-9 was observed at higher ARA inclusion levels. Regarding the DHA, a significant decrease of this FA was observed along with the dietary ARA increase, while the EPA showed a random pattern with a general increase. Interestingly, the ARA deposition decreases at a constant rate along with the ARA increase in the dietary treatments.

### 3.3. Gene Expression

The expression of lipid-relevant genes according to different ARA inclusion levels is presented in Figure 1. The genes *acadvl* and *cpt1a* were significantly highly expressed ($p < 0.05$) in the fish fed with the highest ARA enrichment (1.4 %) in comparison with the lower dietary inclusion levels (0.4 and 0.9%) and the Control. Further, the *fas* and *igf1* expression were higher in fish fed the 0.4% diet decreasing toward the ARA level. Fish from Control treatment and 1.4% presented the lowest expression for these genes at 1.4% treatment. The expression of *ppara* decreased as the ARA was increased in the diet, where the highest expression level was found in the fish-fed Control diet. The presence of ARA modified the *alox5* expression, in accordance to ARA enrichment. No differences were found in the *elovl* gene expression among the different treatments.

### 3.4. Cortisol and Serum Parameters

The total protein serum, glucose, and cortisol levels are presented in Figure 2. The cortisol levels were higher in the Control fish and 1.4% groups, mainly compared to those from 0.9%. Glucose levels resulted in a random pattern with the lowest levels in Control and 0.9% fish, with a higher level in the fish of 0.4 and 1.4% treatments. Serum total protein directly responds to the presence of ARA in the diet, with a significant increase in fish of all dietary treatments enriched with ARA compared to the Control treatment. There was no difference in serum total protein between the fish fed the different experimental treatments enriched with ARA.

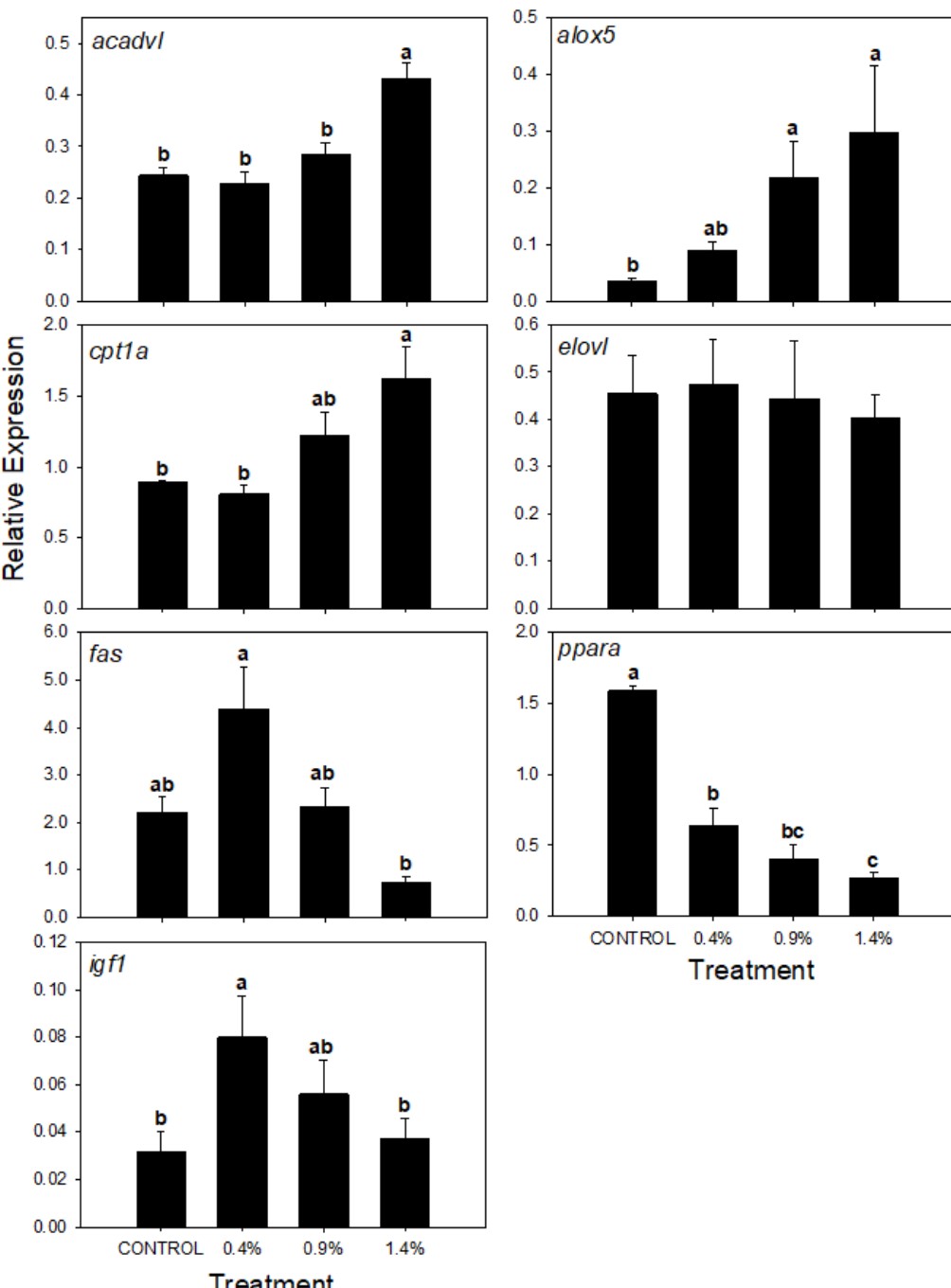

**Figure 1.** acyl-CoA dehydrogenase very long chain (*acadvl*), arachidonate 5-lipoxygenase (*alox*), carnitine O-palmitoyltransferase 1 (*cpt1a*), fatty acid elongase 5 (*elovl*), fatty acid synthase (*fas*), proliferator activated receptor alpha (*ppara*) and insulin-like growth factor (*igf*) relative expression in *S. dorsalis* liver, fed with the different experimental diets. Different letters represent significantly different values ($p < 0.05$) within the same treatment ($n = 6$).

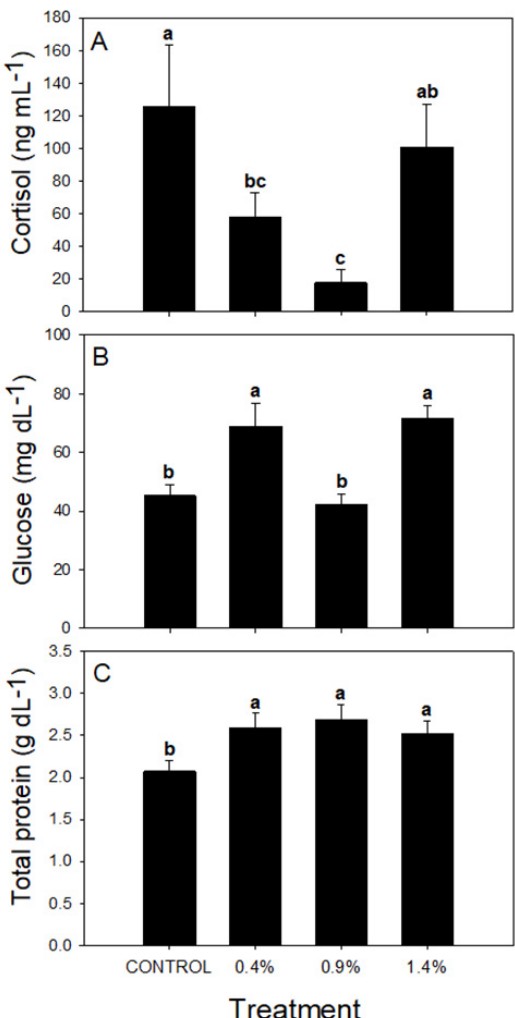

**Figure 2.** Serum cortisol (**A**), glucose (**B**), and total protein (**C**) levels in *S. dorsalis* fed with different experimental diets. Different letters represent significantly different values ($p < 0.05$) between different treatments ($n = 6$).

## 4. Discussion

In the present study, the inclusion of an alternative lipid source (beef tallow) supplemented with ARA was well accepted by the juveniles of *S. dorsalis*. It resulted in consistent growth, survival, and overall performance compared to fish fed diets containing FO. With the high prices and low availability of FO in the world market, alternative lipid sources that are more available and sustainable are essential for manufacturing marine aquafeeds since marine fish, especially carnivorous species, are the most reliable in marine inputs [39]. According to Rombenso et al. [4], such as DHA, ARA is an important LC-PUFA for the adequate development of California Yellowtail juveniles. Therefore, their inclusion in FO-free diets for this species must always be considered. Despite this consistent performance, a gradual increase in ARA levels did not result in better performance (growth and survival) as observed in juveniles of other marine fish species such as *Morone saxatilis* [21] and *Rachycentron canadum* [18]. However, it is essential to mention that the achievement improvement in the performance in those previous studies was at sub-optimal temperature conditions. In this experiment, *S. dorsalis* juveniles were kept at optimal temperature (25 °C), suggesting an effective and positive response of ARA when marine fish are kept at sub-optimal temperatures.

A significant difference observed in the performance results was the higher condition factor in the fish-fed Control diet than those fed ARA diets. This difference can be correlated

with the high *igf1* gene expression. IGF-1 plays an essential role in regulating development and somatic growth, mainly by mediating growth hormone actions since it regulates protein and lipid metabolism [39]. The *igf1* expression resulted in significant differences, where the 0.4% diet showed a significant increase in expression, with a trend to decrease as ARA increases in the diet. Liu et al. [40] reported the same trend in the increase in *igf1* expression when *Sillago sihama* juveniles were fed with the same proportion of lipids and proteins as in 0.4% ARA. Another possible evidence that can justify the higher condition factor in fish from the Control group was the trend to a higher VSI, which means higher fat deposition on the viscera, compared to fish from other experimental groups. Higher HSI and VSI in fish constantly results in changes in the body shape, consequently affecting condition factor [41–43]. However, future studies aiming to understand better this relationship between lipid deposition and *igf1* expression influencing differential body shape are needed.

Regarding the FA profile, the FA deposition differences were observed in the liver and muscle tissue. By changing the LC-PUFAs ratios, differences in lipid metabolism are encountered. In several marine fish species, the FA profile of the tissues directly reflected the dietary FA composition [21,30,44,45]. However, in this study, the total SFA levels in the liver decreased with the increase of ARA in the dietary treatments, while it was not observed in muscle tissue. Marine fish species generally preferentially catabolize SFA to produce biochemical energy through the intermediate metabolism [4,46–48]. Such as previously mentioned by Tocher [1], these findings suggest that California Yellowtail used SFA preferentially over MUFA to generate energy via β-oxidation processes at this life stage and culture conditions.

According to Bell and Sargent [5], ARA directly influences processes related to the catabolism of specific FA. Interestingly, this study observed a reduction of ARA and especially DHA deposition in the muscle according to the increase of ARA in the experimental diets. These results agreed with the high *acadvl* and *cpt-1* expression in fish fed with high ARA levels. These genes encode for membrane transporters that transport FA from mitochondrial cytosol to the mitochondrial matrix, with *acadvl* transporting especially LC-PUFA [49]. Similarly, in a previous experiment with *Oncorhynchus mykiss* juvenile, Morash et al. [50] found a significant increase in CPT I mRNA expression in red muscle, liver, and adipose tissue in fish fed with a diet containing higher PUFA levels. The low DHA in muscle and, inversely, the high expression of *acadvl* and *cpt1* in fish fed higher ARA levels (1.4%) suggest induction of LC-PUFA oxidation (mainly DHA) by the ARA excess in the diet. Despite this study being performed with small fish, ARA impaired fish quality since DHA is highly relevant to human nutrition and health. However, a fish´s FA retention/catabolism process is precise and modulated by several variables, mainly life stage and temperature [1]. Thus, future studies using fish larger than these are needed to elucidate this question.

The inclusion of ARA did not affect the DHA levels of the liver, which were no different than the levels of fish fed the Control diet. There was also no difference in levels of EPA between the fish fed different ARA levels. This profile could be related to LC-PUFA (EPA and ARA) in eicosanoid syntheses. In teleost species, the eicosanoid syntheses occur mainly in the liver [1], and several studies with marine fish species showed that n-6 and n-3 LC-PUFA compete for the same enzymes to synthesize eicosanoids [1,5,44,51]. These previous studies supported that the syntheses of these important lipid mediators are commonly related to the proportion of its precursors (EPA or ARA) in the diet [5,52].

Hixson et al. [53] found more significant correlations among biosynthesis-related genes and muscle tissue FA than with dietary FA, suggesting that the metabolism and storage of muscle tissue FA directly influenced the expression of hepatic genes involved in this LC-PUFA biosynthesis. Such as PPAR-α, which acts as a sensor of FA, in particular n-3 PUFA [34,54,55]. The genes *cptl1a* and *ppara* are directly related to FA catabolism, and their expression could reflect the occurrence of ß-oxidation [34,54,55]. In this study, *ppar-α* was upregulated in fish fed the Control diet compared to fish fed the diets enriched with ARA.

Different diet compositions and, consequently, different FAs result in different effects on *ppar-α* activation or induction in *ppar-α* target genes [56]. The present study results are similar to those found in the literature. The Control diet presented higher levels of EPA than ARA-enriched diets, which explains the upregulated expression of this gene in fish from the Control group. Similar results were observed by Tian et al. [57] in experiments with *Ctenopharyngodon idellus* in which EPA and DHA levels in the hepatopancreas and whole body were negatively correlated with dietary ARA content. The gene *cpt1a* has a higher affinity to MUFA (and EPA), while *ppara* only to EPA. Previous studies on teleost revealed that the *cpt1a* and *ppara* expression connection is controversial. In rainbow trout (*Oncorhynchus mykiss*) [50] and juvenile turbot (*Scophthalmus maximus*) [58], there was no observed apparent connection in the expression patterns of these two genes. While, for olive flounder (*Paralichthys olivaceus*) [59], juvenile black seabream (*Acanthopagrus schlegelii*) [13], and even in *S. dorsalis* fed with different SFA:MUFA ratios [34] a clear connection was noticed. The expression of *cpt1a* was increased with higher levels of ARA, mainly at 1.4%. Because *cpt1a* activates and transports LC-PUFA into the mitochondrial matrix for catabolism [60], increased dietary levels of ARA in the diet likely resulted in either LC-PUFAS β-oxidation rather than storage or synthesis [53]. In particular, they affect the EPA accumulation in the liver, showing an apparent EPA reduction while ARA was accumulated.

Fatty acids biologically active as DHA and ARA are commonly and selectively deposited in the lipolytic and lipogenic tissues such as muscle and liver. In contrast, as previously mentioned, SFA and MUFA are catabolized to provide metabolic energy by β-oxidation processes [1,44,61]. However, the intensity of these processes is highly dependent on the species-specific characteristics, besides other variables, such as rearing conditions [1,21,46,52]. Furthermore, changes in dietary n-3/n-6 ratios can directly modify EPA/ARA ratios in fish tissues [44]. The results suggest that even if ARA is essential for teleost fish, in excess could be prejudicial.

In fish, arachidonate 5-lipoxygenase (*alox5*) is responsible for altering ARA into a bioactive lipid mediator often associated with inflammation [33,62]. *alox5* synthesizes leukotrienes which mediates leukocyte migration from the blood to inflamed tissues, resulting in aggregation, superoxide generation, and mobilization of neutrophils [33]. The expression of the *alox5* gene directly responds to the presence of LC-PUFA in the diet [33,62,63]. As ARA rose in the diet, *S. dorsalis* showed a dose-response increase in *alox5* expression, similar to that found in *L. calcarifer*, where no differences in growth were obtained despite increasing *alox5* expression [63]. Wang et al. [62] reported that the levels of *alox5* are markedly reduced when the levels of PUFAs exceed a threshold of concentration in the diet for *L. crocea*, with concomitants reduction in the immune capacity of the organisms. Therefore, the concentration of ARA in the different experimental diets was appropriate for the organisms to respond appropriately to environmental or biological factors [21,33].

The enzyme fatty acid synthase (FAS) catalyzes the de novo synthesis of fatty acids, which catalyzes the entire pathway of palmitate synthesis from malonyl-CoA [57,64]. The *fas* expression was upregulated in the liver of fish fed 0.4%, contrasting with the significant reduction of expression levels in the highest concentration of ARA (1.4%), possibly due to the decrease of SFA levels, especially on C16:0, on this diet. The same trend to reduce *fas* expression as ARA increase was observed in *C. idellus* [57]. Given the physiological importance of 16:0 in phospholipids composition [65], in preferential FA oxidation order of SFA [1] from shorter to longer chains, *fas* expression was possibly upregulated to compensate for the lower SFA levels in the presence of ARA. Several studies with different fish species have shown more oxidation of SFA and MUFA when offered in excess, consequently preserving the LC-PUFA mainly in the muscle [4,44,47,63,66,67]. Moreover, high dietary PUFA, as usually occurs in fish-oil-based diets, may not be the most efficient practice, as the excess of these FA is commonly oxidized [47,63,68]. The fatty acid elongase (*elovl*) expression was not modified by the presence of ARA in the diet. These results contrast with a study performed with *Solea senegalensis*, where ARA induced a

marked dose-dependent upregulation in the expression of *elovl* [48]. However, it is essential to mention that this previous study found significant differences in *elovl* expression due to the high variation in the levels of specific fatty acids in the diets, especially 18:3n-3 and 18:2n-6, that are a substrate for this enzyme.

Several studies evaluate the effects of dietary FA composition on stress response [5,25,45,52,69–71]. Blood parameters such as cortisol concentration, total protein concentration, and hemoglobin are used as health condition indexes of fish for illness diagnostics or stress indicators. Therefore, when fish experience a stressor, blood parameters are usually altered [72–74]. The higher serum protein levels observed in treatment groups compared to fish from the Control cannot be attributed to the ARA levels in the diets since the Control diet had the same ARA level compared to the 0.4% diet (2.39% and 2.21%, respectively). Moreover, according to Chee et al. [20], ARA affects cortisol and stress response in fish. Our results show a tendency to reduce cortisol and glucose levels in an average concentration of ARA (0.9%). The same beneficial cortisol and modulated stress response were observed in *S. senegalensis* [27] and *S. aurata* [25]. Nevertheless, an adverse effect was observed for these parameters when fish was fed with higher levels of ARA (1.4%). The same dose-dependent effect was observed in *C. idellus*, where blood parameters are adversely affected by ARA levels higher than 0.30% [57]. Moreover, excessive dietary ARA has been reported to cause unusual or lack of pigmentation issues in early developmental stages [28,29]. Therefore, our results suggest that ARA has beneficial effects in modulating the immunological response. However, it is necessary to be cautious in the inclusion levels of ARA in diets to avoid a counter-productive effect.

## 5. Conclusions

This study demonstrates the feasibility of using FO-free diets adequately supplemented with DHA, EPA, and ARA to California yellowtail at this life stage and under these specific conditions. In general, FO-free diets did not affect the growth performance of *S. dorsalis*. Compared to the other treatments, 0.9% ARA supplementation improved feed efficiency and stress response, positively modulating the expression of lipid metabolism-related genes. ARA is important for several physiological processes and benefits the immune status of *S. dorsalis* juveniles. However, in excess, ARA could be prejudicial to a critical concern of the aquaculture: desirable levels of DHA and EPA in the fillets.

**Author Contributions:** B.C.A.: conceptualization, data analysis, writing (original draft). A.K.S.: conceptualization, trial conduction, laboratory analysis. V.H.M.: data analysis, writing (original draft), writing (review). A.T.: trial conduction, laboratory analysis. O.B.D.R.-Z.: laboratory analysis, writing (review). M.T.V.: conceptualization, project administration, data analysis, writing (original, review). J.A.M.-S.: conceptualization, project administration, laboratory analysis, data analysis, writing (original, review). All authors have read and agreed to the published version of the manuscript.

**Funding:** This research was funded by Maricultivos Baja Sel S.A. de C.V. (Ensenada, Mexico) and Fundacão de Amparo a Pesquisa do Estado de São Paulo (FAPESP), project number: 2018/13000-2). The project was financed by UABC (22a/403/1/C/10/22).

**Institutional Review Board Statement:** All procedures in the present study were conducted and authorized according to the UABC animal ethics committee (protocol UABC-IIO 00034/21).

**Data Availability Statement:** Not Applicable.

**Acknowledgments:** We thank Maricultivos Baja Sel S.A. de C.V. (Ensenada, Mexico) for providing the experimental animals and Fundacão de Amparo a Pesquisa do Estado de São Paulo (FAPESP) for the postdoctoral fellowship of Bruno Araujo (project number: 2018/13000-2). This project was also financed by UABC (22a/403/1/C/10/22). We also appreciate the work of Karla Arce, Uriel Tolama-Sosa, and Ivan Abraham Nolasco-Molina.

**Conflicts of Interest:** The authors report no conflict of interest. The authors alone are responsible for the content and writing of the paper.

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
