# Peer review of "Dietary Arachidonic Acid (20:4n-6) Levels and Its Effect on Growth Performance, Fatty Acid Profile, Gene Expression for Lipid Metabolism, and Health Status of Juvenile California Yellowtail (Seriola dorsalis)"

_fishes, doi:10.3390/fishes7040185_

Round 1

Reviewer 1 Report

This study, while not finding statistically significant differences among treatments in production parameters, could be of interest to those engaged in the culture of this species. The authors should consider the following and adjust their MS accordingly.

- I cannot find any specific information about the survival of the fish throughout the study, although the authors mention it (Line 325). The authors should indicate what the survival was for each treatment, including statistical analysis.

- The Introduction should include something about the test species, adding context why this research is important. 

- Table 4. The final condition factor CF of the fish in the Control is much different than that of those in the experimental treatments, despite similar mean weights among all treatments. Is this value correct? From my back-calculations, the length of the Control fish was about 11 cm and that of the experimental fish was about 13.6-13.8 cm. Why such a large difference in length but not weight gain?

- The calculations of weight gain (%) is erroneous. The equation of how it is measured (Line 153) is wrong (the weight gain needs to be divided by the initial weight). The values (Table 4) are wrong; the final mean weight of fish in all the treatments is more than double that of the initial mean weight, so weight gain as a percent of initial must exceed 100%. But, see next comment.

- Given that the SGR and the initial and final weights are given (Table 4), there is no need to provide weight gain data. Weight gain as a percent of initial weight is generally not a good metric of growth for comparison purposes, because it is overly affected by small variation in initial weight. SGR, and perhaps even better Thermal Growth Coefficient TGC, are more accepted measures of growth response. 

- Table 1. The authors should describe better what ARA-enriched oil is. It is > 40% ARA, but what is it is derived from? What is the composition, including fatty profile, of the other 60%?

- The authors used ANOVA to analyze their data, but given the graded increase in levels of ARA, regression analysis could provide more meaningful insight. The authors mention a "tendency to increase with higher inclusion levels of ARA" for weight, but it is unclear as to how meaningful this tendency is. Regression analysis could be useful here, although with only 3 levels under consideration, it's hard to make many inferences. See next comment for post-hoc test comment.

- The authors should explain why they used a rank test (Line 238) to examine differences in treatments instead of a regular test. While I believe regression analysis is a good approach (see previous comment), with regards to a range test, I suggest that Dunnet's (or similar) test, which compares multiple treatment means against a control, could be most useful here. 

- Lines 388-391 and 441-444 are identical, which causes confusion for the reader. I'm not sure of a fix, but the authors should carefully consider the two instances of this sentence and rephrase or remove one of them.

Author Response

Reviewer 1

1 - This study, while not finding statistically significant differences among treatments in production parameters, could be of interest to those engaged in the culture of this species. The authors should consider the following and adjust their MS accordingly.

Response: We appreciate your positive feedback. Your comments and suggestions were critical to improving the quality of our manuscript.

2 - I cannot find any specific information about the survival of the fish throughout the study, although the authors mention it (Line 325). The authors should indicate what the survival was for each treatment, including statistical analysis.

Response: Sorry for that. As requested, we have included the survival rate in Table 4.

3 - The Introduction should include something about the test species, adding context why this research is important. 

Response: We agree, and as suggested, a sentence about this species was included (lines 78-84).

4 - Table 4. The final condition factor CF of the fish in the Control is much different than that of those in the experimental treatments, despite similar mean weights among all treatments. Is this value correct? From my back-calculations, the length of the Control fish was about 11 cm and that of the experimental fish was about 13.6-13.8 cm. Why such a large difference in length but not weight gain?

Response: We agree that this is an unusual result. We have carefully double-checked the individual weight and fork-length data, which seems to be correct. Despite being difficult to explain, some pieces of evidence could corroborate this result. First of all, a not significant trend was observed to higher HSI and mainly VSI in fish from the control group. Additionally, the expression of igf1 (which modulates the somatic growth in fish) followed an inverse profile compared to CF results. We have included this information in the manuscript (lines 346-360)

5 - The calculations of weight gain (%) is erroneous. The equation of how it is measured (Line 153) is wrong (the weight gain needs to be divided by the initial weight). The values (Table 4) are wrong; the final mean weight of fish in all the treatments is more than double that of the initial mean weight, so weight gain as a percent of initial must exceed 100%. But, see next comment.

Response: We agree, and as requested, the weight gain calculation was removed from Table 4 and Material and Methods.

6 - Given that the SGR and the initial and final weights are given (Table 4), there is no need to provide weight gain data. Weight gain as a percent of initial weight is generally not a good metric of growth for comparison purposes, because it is overly affected by small variation in initial weight. SGR, and perhaps even better Thermal Growth Coefficient TGC, are more accepted measures of growth response. 

Response: We agree that SGR is the most adequate index to compare growth performance, and as requested we removed the weight gain calculation from Table 4 and Material and Methods.

7 - Table 1. The authors should describe better what ARA-enriched oil is. It is > 40% ARA, but what is it is derived from? What is the composition, including fatty profile, of the other 60%?

Response: As requested, additional information and fatty acid composition of the ARA-enriched oil were included in line 141, and Table 2.

8 - The authors used ANOVA to analyze their data, but given the graded increase in levels of ARA, regression analysis could provide more meaningful insight. The authors mention a "tendency to increase with higher inclusion levels of ARA" for weight, but it is unclear as to how meaningful this tendency is. Regression analysis could be useful here, although with only 3 levels under consideration, it's hard to make many inferences. See next comment for post-hoc test comment.

Response: We entirely understand the reviewer's concern regarding the statistics. We also agree that regression should be the adequate method in nutritional trial testing a gradient inclusion of ingredients. However, since the control diets with fish oil compared to the experimental diets containing beef lard plus cholesterol and DHA, the Control cannot be included in the regression. Therefore, as stated by the reviewer, we also agree that three points cannot be appropriate to analyze this experiment. For this reason, we opted to keep the analysis stats using ANOVA.

9 - The authors should explain why they used a rank test (Line 238) to examine differences in treatments instead of a regular test. While I believe regression analysis is a good approach (see previous comment), with regards to a range test, I suggest that Dunnet's (or similar) test, which compares multiple treatment means against a control, could be most useful here. 

Response: Sorry for our mistake. It was performed ANOVA followed by a regular test (Tukey). Aiming to clarify the stats methodology used, we changed and complemented this information in the manuscript (lines 239-242). In addition, we agree that Dunnet’s test could be adequate to analyze treatments vs a control group. However, in our opinion, since the control group had 2.39% of ARA (from fish meal and fish oil) in their composition, Dunnet’s is not entirely adequate. Thus, Tukey could be more appropriate since it compares every means with every other mean, including the control group.  We chose to use this posthoc test based on similar previous studies published by our and other research groups. Please, find some examples below:

https://doi.org/10.1016/j.aquaculture.2019.734245

https://doi.org/10.1016/j.aquaculture.2020.735939

https://doi.org/10.1016/j.aquaculture.2020.735207

https://doi.org/10.1016/j.aquaculture.2019.734644

10 - Lines 388-391 and 441-444 are identical, which causes confusion for the reader. I'm not sure of a fix, but the authors should carefully consider the two instances of this sentence and rephrase or remove one of them.

Response: Sorry for our mistake. The information was removed from lines 441–444.

Reviewer 2 Report

The manuscript by Araujo et al. evaluated incremental doses of 20:4n-6 for Seriola dorsalis. The manuscript is well written, and it appears to be properly conducted. Below you can find some questioning and suggestions:

L153: Why is weight gain not divided by the initial body weight? The percentage of weight gain is usually normalized by the initial body weight (NRC, 2011).

L162: Specify the acronym UABC

163: typo? Purposes?

L183: Typo for temperature degree (at this line and throughout the manuscript), superscript -1 at this line and throughout the manuscript

L195-196: This sentence reads weirdly; it may require revision.

L224-227: An additional set of blood samples was taken? Or is it repeating L165-167?

L236-240: Was the data validated for normal distribution and homogeneity of variance? Please add this information to the Statistical analysis segment.

L396: diet, not diets.

L466-484: What could be driving a higher protein concentration in the serum by the ARA supplementation?

L481-482: This may be a little farfetched to claim when only testing glucose, cortisol, and protein in the serum and the expression of one inflammatory gene. I suggest rephrasing it.

Author Response

Reviewer 2

11 - The manuscript by Araujo et al. evaluated incremental doses of 20:4n-6 for Seriola dorsalis. The manuscript is well written, and it appears to be properly conducted. Below you can find some questioning and suggestions:

Response: We appreciate your positive feedback. Your comments and suggestions are critical to improving the quality of our manuscript.

12 - L153: Why is weight gain not divided by the initial body weight? The percentage of weight gain is usually normalized by the initial body weight (NRC, 2011).

Response: Sorry for our mistake. As requested by reviewer #1, we removed the weight gain from Table 4, since SGR and FCR are the most adequate variables to evaluate fish growth performance.

13 - L162: Specify the acronym UABC

Response: Included as requested (lines 168-169).

14 - 163: typo? Purposes?

Response: Changed as requested

15 - L183: Typo for temperature degree (at this line and throughout the manuscript), superscript -1 at this line and throughout the manuscript

Response: Changed as requested

16 - L195-196: This sentence reads weirdly; it may require revision.

Response: Changed as requested

17 - L224-227: An additional set of blood samples was taken? Or is it repeating L165-167?

Response: Sorry for our mistake. You are right, the information repeated was removed from lines 224-227.

18 - L236-240: Was the data validated for normal distribution and homogeneity of variance? Please add this information to the Statistical analysis segment.

Response: Sorry for our mistake. The data was tested previously for normality and homogeneity. This information is now included in the manuscript (lines 239-242).

19 - L396: diet, not diets.

Response: Replaced as requested

20 - L466-484: What could be driving a higher protein concentration in the serum by the ARA supplementation?

Response: Unfortunately, the reasons behind the higher protein concentration observed in groups 0.4, 0.9, and 1.4% are not clear yet. However, it should possibly not be related to the ARA levels since the Control group had 2.39% of this same fatty acid in their composition. We have included this information in the manuscript (lines 478-480).

21 - L481-482: This may be a little farfetched to claim when only testing glucose, cortisol, and protein in the serum and the expression of one inflammatory gene. I suggest rephrasing it.

Response: We agree and rephrased this sentence as requested.
